# Blood Biomarkers of Alzheimer’s Disease and Cognition: A Literature Review

**DOI:** 10.3390/biom14010093

**Published:** 2024-01-11

**Authors:** Greta Garcia-Escobar, Rosa Maria Manero, Aida Fernández-Lebrero, Angel Ois, Irene Navalpotro-Gómez, Victor Puente-Periz, José Contador-Muñana, Isabel Estragués-Gazquez, Albert Puig-Pijoan, Joan Jiménez-Balado

**Affiliations:** 1Hospital del Mar Medical Research Institute, 08003 Barcelona, Spain; ggarciaescobar@psmar.cat (G.G.-E.); rmanero@psmar.cat (R.M.M.); afernandezlebrero@psmar.cat (A.F.-L.); inavalpotro@psmar.cat (I.N.-G.); vpuente@psmar.cat (V.P.-P.); josemiguel.contador.munana@psmar.cat (J.C.-M.); iestragues@psmar.cat (I.E.-G.); apuig@psmar.cat (A.P.-P.); joan.balado@gmail.com (J.J.-B.); 2Neurology Department, Hospital del Mar, 08003 Barcelona, Spain; 3Department of Health and Experimental Sciences, Universitat Pompeu Fabra, 08003 Barcelona, Spain; 4Department of Medicine, Universitat Autònoma de Barcelona, 08193 Barcelona, Spain

**Keywords:** Alzheimer’s disease, cognitive impairment, blood biomarkers, cognitive functions, cognition

## Abstract

Recent advances in blood-based biomarkers of Alzheimer’s Disease (AD) show great promise for clinical applications, offering a less invasive alternative to current cerebrospinal fluid (CSF) measures. However, the relationships between these biomarkers and specific cognitive functions, as well as their utility in predicting longitudinal cognitive decline, are not yet fully understood. This descriptive review surveys the literature from 2018 to 2023, focusing on the associations of amyloid-β (Aβ), Total Tau (t-Tau), Phosphorylated Tau (p-Tau), Neurofilament Light (NfL), and Glial Fibrillary Acidic Protein (GFAP) with cognitive measures. The reviewed studies are heterogeneous, varying in design and population (cognitively unimpaired, cognitively impaired, or mixed populations), and show results that are sometimes conflicting. Generally, cognition positively correlates with Aβ levels, especially when evaluated through the Aβ_42_/Aβ_40_ ratio. In contrast, t-Tau, p-Tau, Nfl, and GFAP levels typically show a negative correlation with cognitive performance. While p-Tau measures generally exhibit stronger associations with cognitive functions compared to other biomarkers, no single blood marker has emerged as being predominantly linked to a specific cognitive domain. These findings contribute to our understanding of the complex relationship between blood biomarkers and cognitive performance and underscore their potential utility in clinical assessments of cognition.

## 1. Introduction

Alzheimer’s disease (AD) is the main cause of dementia, and the World Health Organization recognizes it as a public health priority. The main clinical manifestations of AD are the progressive impairment of memory and other cognitive domains such as language, attention, executive functions (EF), and visuospatial abilities [1], which lead to a decline in daily life activities, usually accompanied by neuropsychiatric symptoms. Objective cognitive and functional impairment is assessed through neuropsychological evaluations, which involve standardized tests across multiple cognitive domains. The obtained scores are then compared to normative scales, considering relevant demographic factors such as the patient’s age and education. This comparison helps to quantify the deviation of the patient’s performance from the expected norm [2]. Thus, neuropsychological assessment remains essential in the diagnostic process of cognitive decline.

Alzheimer’s pathology is defined by the accumulation of extraneuronal amyloid plaques, composed of abnormal beta-amyloid proteins, and intraneuronal neurofibrillary tangles, characterized by misfolded tau proteins, in the brain [3]. Initially, the diagnosis of AD was based on the stage of dementia explained by these cognitive, functional, and behavioral symptoms [4], but the advances in the biomarker field have enabled researchers to detect the pathophysiological characteristics of the disease in vivo and contribute to a more accurate diagnosis criterion for AD. AD biomarkers are included in the current criteria for ADby National Institute on Aging and Alzheimer’s Association, commonly used both in clinical [5,6] and research settings [7]. Cerebrospinal fluid (CSF) can reflect pathological changes in brain tissue due to its direct contact. Although CSF is considered to be an optimal measure, it requires an invasive technique of acquisition (lumbar puncture), and this underlies the need for less invasive techniques [8,9]. During recent years, novel blood-based biomarkers of AD have been developed in a research context. Plasma biomarkers of AD are promising due to their less invasive and cost-effective extraction methods [10]. Further studies are needed to explore the applicability of blood-based AD biomarkers in the clinical practice of memory units.

We aim to conduct a descriptive review of the latest literature spanning the last five years (2018–2023), focusing on novel blood-based biomarkers of AD such as amyloid-β (Aβ) and p-Tau, markers of neurodegeneration, including t-Tau and Neurofilament Light (NfL), as well as a glial activation marker, Glial Fibrillary Acidic Protein (GFAP), exploring their associations with cognitive measures.

## 2. Materials and Methods

### 2.1. Study Design

#### 2.1.1. Search Strategy and Selection Criteria

We conducted a comprehensive literature search on PubMed using the following strategy: ‘Plasma biomarkers’ AND ‘Alzheimer’s disease’ AND ’Cognitive change’. This search specifically targeted articles published within the last five years (2018–2023).

#### 2.1.2. Inclusion and Exclusion Criteria

Our inclusion criteria focused on studies that investigated blood-based biomarkers of AD and the related neurodegeneration and glial activation (Aβ, p-Tau, t-Tau, NfL, and GFAP) in human subjects. We selected studies that utilized recognized cognitive assessments and presented findings from both cross-sectional and longitudinal study designs. The study populations were diverse, including samples with both cognitively impaired (CI) and cognitively unimpaired (CU) subjects (mixed population), as well as those exclusively comprising either CI or unimpaired individuals. We excluded studies not directly addressing AD, lacking clear cognitive outcome measures, or based on non-human subjects or in vitro models.

#### 2.1.3. Peer Review Process

All the preselected articles identified through our search criteria were further reviewed by three senior researchers (GGE, AO, and APP). This rigorous review process was implemented to verify the quality and relevance of each study, ensuring the inclusion of only the most pertinent and scientifically robust research in our review.

#### 2.1.4. Biomarkers Measurements

The recent development of advanced technologies has significantly enhanced the detection capabilities for plasma proteins. In the articles included in this review, various techniques and platforms have been employed to determine disease-related proteins with high sensitivity and precision, even at low concentrations. Among these platforms is SIMOA (Single Molecule Array, Quanterix Corporation (Billerica, MA, USA), a digital immunoassay technique which enables the detection of biomarkers at extremely low concentrations in blood samples. This technology is particularly significant for AD research, as it allows for the accurate measurement of neurological biomarkers present in minute quantities, thereby facilitating early detection and monitoring of disease progression. Other notable platforms include MSD (Mesa Scale Discovery, Meso Scale Diagnostics, LLC (Rockville, MD, USA)), LC-MS/MS (Liquid Chromatography–Mass Spectrometry), ELISA (Enzyme-Linked Immunosorbent Assay), AlphaLISA (Amplified Luminescent Proximity Homogeneous Assay, PerkinElmer (Shelton, CT, USA)), and Olink Proteomics (Uppsala, Sweden). The detailed methodologies for each platform are outlined in the cited articles.

#### 2.1.5. Data Extraction and Synthesis

Data were extracted from the selected articles, focusing on the relationship between blood-based biomarkers and cognitive measures in AD. We emphasized studies using established neuropsychological tests across cognitive domains. Global cognition was primarily assessed using the Mini-Mental State Examination (MMSE) [11] and the Montreal Cognitive Assessment [12]. The memory domain was evaluated through tasks like the Auditory Verbal Learning Test (AVLT) [13] and the Free and Cued Selective Reminding Test [14], while visuospatial abilities were often measured using tests like the Clock Drawing Test (CDT) [15] (historical review). Language abilities were examined with tests such as semantic fluency (SF) or the Boston Naming Test [16], and the attention/EF domains were assessed using tests like verbal span, the Trail Making Test (TMT) [17], and the Stroop Color and Word Test [18]. This inclusion helps us to underline the ongoing validity of these neuropsychological tests in both routine clinical practice and research settings.

#### 2.1.6. Presentation of Findings

The findings are synthesized descriptively, highlighting the correlations between blood-based biomarkers and cognitive measures in the AD continuum. Summary tables (Table 1 and Table 2) are included to provide a clear and concise overview of the observed associations of the main blood-based AD biomarkers: plasma Aβ and p-Tau and t-Tau measures. To enhance the clarity and accuracy of our review, we have implemented a systematic approach for assigning directional arrows in the correlation tables, which reflect the strength and nature of the correlations reported in the studies. This evaluation process was meticulously carried out by two senior neurologists and one neuropsychologist, ensuring a thorough and expert analysis of each study’s findings. A single upward arrow (↑) indicates a positive correlation identified through univariate analyses or in studies with smaller sample sizes. Two upward arrows (↑↑) denote a stronger positive correlation, typically observed in studies with larger sample sizes or multivariate analyses. For negative correlations, where higher biomarker levels correspond to poorer cognitive performance or faster cognitive decline, a single downward arrow (↓) is used for univariate analyses or smaller studies, and two downward arrows (↓↓) are employed for stronger negative correlations identified in larger sample sizes or multivariate analyses.

### 2.2. Blood-Based Aβ

The accumulation of Aβ peptides, associated with AD pathophysiology, becomes detectable up to 20–30 years before the onset of clinical dementia [19]. Various studies have evaluated the accuracy of blood-based Aβ biomarkers in detecting Aβ pathology, using brain amyloid PET scans and/or CSF Aβ biomarkers as the gold standard [20,21,22]. While blood-based measures of Aβ offer valuable insights, it is important to note that their reliability tends to be poorer compared to Aβ measures in CSF. Additionally, studies have found lower plasma Aβ levels in individuals with AD-type dementia compared to CU individuals and other diagnostic groups [23], although these blood-based measures tend to show less variation between groups than those in CSF [24]. Furthermore, plasma Aβ levels have demonstrated the ability to differentiate AD-type dementia from mild cognitive impairment (MCI) patients [25], with MCI being classically defined as cognitive impairment without meeting dementia criteria [26]. On the other hand, some studies have reported that the Aβ42/40 ratio in plasma exhibits a better correlation with Aβ pathology than individual Aβ42 and Aβ40 measures alone [27,28].

#### Association between Blood-Based Aβ and Cognition

Some cross-sectional studies in mixed samples, including CU and cognitive impairment subjects, have found a significant positive association between the plasma Aβ_42_/Aβ_40_ ratio or plasma Aβ_42_ and MMSE, as well as with verbal memory tests such as the AVLT and Story Recall; and a visuospatial task, the CDT [29,30]. Moreover, a longitudinal study involving mixed samples found that lower baseline plasma Aβ_42_/Aβ_40_ ratio levels were associated with poorer performance in a verbal memory task (Word List Learning) and an attention/EF composite score, which included semantic fluency, letter fluency (LT), digit span, and visual span, and that a steeper cognitive decline was observed over approximately 1.5 years [31]. Conversely, other cross-sectional studies failed to show a correlation between Aβ_42_ and Aβ_42_/Aβ_40_ ratio and the MMSE [32] and an attention/EF task, the TMT [33], in mixed samples. Similarly, other studies with longitudinal data findings did not find an association between baseline plasma Aβ_42_/Aβ_40_ and longitudinal decline in the MMSE [32,33,34].

Regarding studies involving only CI samples, a study conducted by Chen et al. in 2019 [35] revealed a significant positive cross-sectional association in AD patients between plasma Aβ_42_ and attention/EF composite scores. These composite scores included tests such as the Stroop Color–Word Test (SCWT), TMT Part B, and SF, as well as Digit Span (DS) recall. Additionally, they found a correlation with a multi-domain composite score observed in both MCI and AD patients. Furthermore, they observed a significant positive association between the plasma Aβ_42_/Aβ_40_ ratio and memory composite scores among individuals with MCI during a 3-year follow-up period. These composite scores included verbal memory tests like the AVLT and Story Recall, as well as a visual memory task, the Modified Rey–Osterrieth Complex Figure Test (MROCFT), and attention/EF scores (forward/backward DS recall). Consistent with these findings, Tsai et al. (2020) [36] demonstrated a significant negative correlation between plasma Aβ_42_ levels and the annual changes in MMSE scores in an amnestic MCI group. However, another study did not find a cross-sectional association or a longitudinal decline between the plasma Aβ_42_/Aβ_40_ ratio and a brief multi-domain cognitive battery, the Addenbrooke’s Cognitive Examination Revised version (ACE-R), in MCI and AD patients [37].

Concerning studies of exclusively CU samples, a cross-sectional study revealed a positive association between CU participants with a subjective cognitive decline (SCD) and a lower plasma Aβ_42_/Aβ_40_ ratio, which were correlated with poorer performance on a novel episodic memory task called the Face–Name Associative Memory Exam [38]. Additionally, in a recent longitudinal study by Cullen et al. (2021) [39], they predicted that CU individuals with SCD would experience a more pronounced decline in scores on the Preclinical Alzheimer’s Cognitive Composite (PACC) test over a period of approximately 4 years. Other studies have corroborated this trajectory in multi-domain composite cognitive scores, which incorporate tasks related to verbal memory (e.g., Free and Cued Selective Reminding Test, FCSRT) and attention/EF (e.g., Digit Symbol Substitution Test, DSST). These studies compared individuals classified as Aβ+ CU and Aβ- CU, based on plasma Aβ_4_2/Aβ_40_ ratio levels. Specifically, Giudici et al. (2020) [40] observed this trend over a follow-up period of 3.9 years, while Aschenbrenner et al. (2022) [41] noted a more rapid decline over an extended follow-up period of approximately 12 years. Similarly, Aβ-positive SCD participants showed a decline in verbal memory (as measured using Verbal Learning Test scores) and in an attention/EF task (as measured by TMT Part B) after 2 years [42]. However, it is worth noting that some studies involving CU samples did not report a cross-sectional association between the plasma Aβ_42_ or Aβ_42_/Aβ_40_ ratios and cognitive composite scores [41] or a longitudinal decline over time on measures such as the MMSE [36,43,44] and the PACC [44] and a multi-domain cognitive composite test comprising Wechsler Adult Intelligence Scale-III (WAIS-III) subtests [45]. In line with this, another recent study found that a significant decline in MMSE scores at the two-year follow-up was not associated with a baseline plasma Aβ_42_/Aβ_40_ ratio [46].

### 2.3. Blood-based Total Tau and Phosphorylated Tau

Tau pathology constitutes the other proteinopathy that characterizes AD pathology. The p-Tau aggregated into neurofibrillary tangles has a precipitant role in the neurodegeneration and cognitive decline in the AD continuum [47]. The literature has shown that plasma t-Tau is increased in AD compared to a control group, but there is a large overlap between normal aging and patients without dementia which [34] prevents it from being considered a diagnostic tool [24,48,49]. On the other hand, phosphorylation at threonine 181 (p-Tau181) is the most widely used p-Tau biomarker in the clinical setting and increases in the early phases of AD [24]. Phosphorylation at threonine 217 and threonine 213 (p-Ttau217, p-Tau231) are other phosphorylated tau species increased in AD, and current assays are focusing on them [50]. It has been reported that plasma p-Tau181 levels are increased in AD versus the controls [51]. A recent study has shown that plasma p-Tau181 and p-Tau217 can differentiate patients with an AD CSF profile (defined by the CSF Aβ_42_/p-Tau ratio) from the non-AD CSF profile group in a memory unit [22]. The literature has reported higher levels of plasma p-Tau181, p-Tau217, and p-Tau231 in AD patients, among CU individuals [9,52], and MCI patients [9,53]. It has also shown that plasma p-Tau217 showed better distinguishing accuracy than plasma p-Tau181 and plasma p-Tau231 in the early stages of AD [9,54] and discriminated AD from other neurodegenerative diseases [55]. Other recent studies also show that plasma p-Tau231 can identify AD patients with other non-AD neurodegenerative diseases [56] and that abnormal levels of plasma p-Tau231 are reached in preclinical AD stages [57]. A recent systematic review reported that plasma p-Tau231 increased from CU to MCI to AD and showed an excellent diagnostic accuracy for asymptomatic Aβ pathology [58].

#### Association between Blood-Based t-Tau and p-Tau and Cognition

Some cross-sectional studies of mixed samples have reported a significant negative correlation between higher plasma p-Tau181 levels and MMSE [29,32,59,60] and Montreal Cognitive Assessment (MoCA) measures [59]. In addition, other studies have found this association with verbal memory tests such as AVLT, Story Recall [30,61], a visuospatial task like CDT [30], and single-domain composites. These composites include the following: memory, as measured by means of the AD Neuroimaging Initiative (ADNI)-Memory [59] visuospatial abilities, assessed through overlapping imaging in MoCA-B, MROCFT copy, and the copy form Stick test [29]; language, as measured via the ADNI-Language [59] test; and naming abilities, as assessed in MMSE and MoCA, or with the Boston Naming Test (BNT), and object naming abilities [29], as well as EF, as measured by ADNI-EF [59]. In addition, longitudinal studies have found various associations regarding p-Tau measures. Higher plasma p-Tau181 levels have been linked to a faster decline in MMSE scores [33,60], in the scores of a multi-domain cognitive battery Disease Assessment Scale–Cognitive Subscale with 13 tasks (ADAS-Cog 13) [34], and in the Dementia Rating Scale (DRS) scores [62]. In contrast, Tsai et al. (2020) [36] did not find an association between plasma p-Tau181 and the decline in MMSE scores. On the other hand, a higher plasma p-Tau217 level has been shown to predict declines and annual changes in MMSE scores [33,63], with a larger effect size observed in the prediction of plasma p-Tau217 compared to p-Tau181 [33]. Furthermore, in the study conducted by Smirnov et al. (2022) [62], higher plasma p-Tau231 has also been associated with a more rapid cognitive decline, as measured by the DRS, over time. Regarding plasma t-Tau measures, higher levels have been also associated with poorer performance on the MMSE [30,32], visuospatial tasks, the Hooper Visual Organization Test (HVOT) [49], some memory tasks such as AVLT, WMS memory subtests, and a word list learning test [31,49,64] as well as attention/EF tasks like the TMT [49] and an EF composite that included SF, LF, and verbal and visual span backwards [31]. Continuing with mixed samples, a study showed that a higher concentration of serum t-Tau was correlated with a longitudinal decline in multi-domain cognitive composite scores, which included MMSE, memory, and attention/EF tasks, over a span of 16 years [65]. Another study also reported longitudinal associations between higher t-Tau levels and declines in memory, attention, and global cognitive scores [64]. On the other hand, some findings have concluded that plasma t-Tau levels did not predict changes in MMSE scores [36].

In samples involving only CI, a cross-sectional study showed that MMSE scores are negatively associated with plasma p-Tau181 in AD patients [43]. Longitudinal data revealed a significant correlation between plasma p-Tau181 concentration and longitudinal declines in MMSE, MoCA, and ADNI battery cognitive composite tests in MCI Aβ+ individuals. Additionally, a significant decrease was observed in ADNI-Memory scores in individuals at the AD dementia stage [59]. A negative association has also been found between plasma p-Tau181 levels and a prospective decline in ADASCog 13 scores [66] as well as with ACE-R slopes [37] in MCI and AD subjects. Furthermore, higher baseline plasma p-Tau217 levels have also been associated with a steeper cognitive decline across language tasks (SF and BNT) and attention/EF tasks (DS backwards, TMT, SCWT, LT, and Design fluency) in individuals with MCI. This association was also observed across memory tasks, the California Verbal Learning Test (CVLT), and the Benson Figure test but only in male MCI subjects [67]. Regarding the plasma t-Tau levels in MCI subjects, a study has reported a decrease in MMSE scores during the first few years [32].

Regarding studies with exclusively CU samples, some longitudinal studies have reported that higher plasma p-Tau181 levels are longitudinally associated with lower PACC scores in CU individuals [66]. A study conducted by Thomas et al. (2021) [68] also showed a faster decline in PACC, CDR-Sum of Boxes (CDR-SOB), and Functional Activities Questionnaire (FAQ) scores, which were related to higher plasma p-Tau181 levels in objectively defined subtle cognitive decline (Obj-SCD) subjects. Moreover, a recent finding showed that elevated baseline plasma p-Tau181 levels significantly predicted a greater decline in a multi-domain cognitive composite score (encompassed by the WAISS-III subtest) over 10 years in an ageing cohort [45]. However, in a recent finding, plasma p-Tau181 levels did not correlate with the MMSE scores [69]. Regarding the plasma p-Tau217 levels in CU participants, a study also identified a significant inverse relationship with longitudinal worsening in the MMSE, PACC, and verbal memory (RAVLT scores) measures [44]. Similarly, Saloner et al. (2023) [67] observed a strong correlation between higher plasma p-Tau217 levels and verbal memory decline (CVLT scores) in female CU individuals. Another finding described a strong association between plasma p-Tau217 levels and annual change in PACC scores in SCD participants [39]. In this line, a recent study has stated that plasma p-Tau217 predicted MMSE and PACC scores decline among Aβ+ CU participants [70]. Concerning t-Tau measures, Baldacci et al. (2020) [71] suggested that changes in plasma t-Tau levels might serve as predictors of longitudinal decline in MMSE scores among individuals with SCD at a 3-year follow-up, but they did not find cross-sectional associations with MMSE and FCSRT scores. Additionally, no correlation was found between serum t-Tau and other memory measures such as word recall, motor function such as Purdue Pegboard Test (PPT) scores, and attention/EF scores assessed using the SCWT and Letter Digit Substitution tests (LDST) [72].

### 2.4. Other Blood-Based Biomarkers

In addition to the core biomarkers of AD pathology (Aβ and p-Tau for amyloid and AD tauopathy, respectively) and t-Tau for AD-related neurodegeneration (despite not being a specific AD biomarker), there are other biomarkers involving concomitant pathological mechanisms, such as plasma NfL for neurodegeneration and Glial Fibrillary Acid Protein (GFAP) for neuroinflammation, that are being commonly assessed in AD patients.

### 2.5. Neuronal Injury: Blood-Based Neurofilament Light

NfL is a component of the neural cytoskeleton, and it is a well-stablished marker of neuroaxonal injury and neurodegeneration [23,73]. Higher levels of NfL are present in multiple neurological conditions such as traumatic brain injury, atypical parkinsonian disorders, and amyotrophic lateral sclerosis, among others [74]. Increased plasma NfL levels have been found in MCI and AD when compared with a control group [75,76,77]. A study also found increased rates according to a positive Aβ status within CU and MCI groups, thus associating plasma NfL also with AD pathology [76]. In addition, this biomarker has also demonstrated the ability to predict the progression of the disease in familial AD [78,79].

#### Association between Blood-Based NfL and Cognition

Some findings have reported a statistically significant cross-sectional negative correlation in mixed samples between higher plasma NfL levels and poorer MMSE scores [29,30,80,81]. Additionally, specific attention/EF scores, such as TMT, DSST, and LT Delis Kaplan Executive Function System subtests, have shown similar correlations [30,64,81,82,83]. Visuospatial tasks (CDT, HVOT), memory tasks (logical memory, AVLT, and visual memory), and language tasks (BNT and SF) have also exhibited such associations [29,30,64,82,83], as well as the multi-domain cognitive composite score [81]. In addition, some longitudinal research has shown that elevated plasma NfL levels were also associated with a faster decline in follow-up assessments in global cognitive scores [62,64,65,84], PACC scores [85], and in the composite scores of memory, EF, language, and visuospatial tasks’ [31,85] DRS scores [62], as well as in everyday functioning, as assessed using the FAQ [85]. In another mixed-sample study, it was reported that MMSE scores, ADAS-Cog-11 scores, and CDR-SOB scores were associated with higher baseline levels and a more rapid increase in plasma NfL, regardless of the diagnostic group [76]. Nevertheless, a study conducted by Mielke et al. (2019) [84] did not find any significant cross-sectional associations between plasma NfL and global cognition scores, nor with memory scores (WMS subtests and AVLT), attention/EF scores (TMT and DSST), visuospatial scores (Picture Completion and Block Design), and language scores (SF and BNT). In this line, another study did not show that plasma NfL was associated with a longitudinal decline in cognition, as assessed through MMSE measures [33].

Regarding CI samples, higher plasma NfL levels were also associated with lower performances in multi-domain cognitive composite scores [81], ACE-R scores [37], attention/EF scores, and memory scores [82,83]. Additionally, correlations were found in language scores such as SF and visuospatial function scores like HVOT [82] in individuals with MCI. Lin et al. (2018) [80] found a trend toward a significant negative correlation between plasma NfL levels and MMSE scores in the MCI group, but the correlation was significant in AD patients. Furthermore, a study conducted in an autosomal dominant AD cohort reported significant negative associations between serum NfL levels and MMSE scores, memory tasks (FSCRT), attention/EF (DS and TMT part B), and language tasks (BNT) in mutation carrier subjects [78]. Some longitudinal data are also available such as those of a study conducted by Moscoso et al. (2021) [66], which reported an association with cognitive decline measured by ADAS-Cog scores in cognitively impaired subjects. In other samples exclusively composed of MCI patients, some studies have shown a relationship between higher plasma NfL and a cognitive decline over time assessed using MMSE [43,76], ADAS-Cog, and CDR-SOB scores [76], ADNI-memory composite and PACC scores [85], as well as global cognitive scores [81]. In AD groups, higher plasma NfL and longitudinal worsening in ADAS-Cog scores have been found [76,86]. Li et al. (2021) [86] also reported that higher baseline plasma NfL was associated with a faster decline in activities of daily living functionality, assessed by the Disability Assessment for Dementia (DAD), from the baseline to 12 months in a sample with mild-to-moderate AD dementia. Another recent study pointed out that a combination of higher plasma NfL levels and changes in MMSE scores is a strong predictor of progression from MCI to AD dementia within 5 years [87]. Nevertheless, another study did not find an association between plasma NfL and cognition assessed by ACE-R slopes in MCI with positive AD biomarkers [37].

When examining CU samples, a cross-sectional association was also found between higher plasma NfL levels and worse multi-domain composite cognitive scores [41,88]. This association has also been reported in relation to attention/EF measures such as SCWT and LDST, as well as motor speed tasks like PPT and memory tests such as the 15-Word Learning Test [83,88]. Additionally, NfL serum levels were found to be associated with a multi-domain cognitive composite score, attention/executive function (SCWT and LDST scores), and motor performance (PPT scores) but not with the memory domain [72]. In CU individuals with SCD, Chatterjee et al. (2018) [89] also observed a significant inverse correlation between plasma NfL and attention/EF composite scores (DS backward, DSST), as well as the multi-domain cognitive score. There is also longitudinal research that has reported a negative correlation between baseline serum NfL levels and the annual change in MMSE scores [90]. Other longitudinal findings have suggested an association between higher baseline plasma NfL levels and a decline over time in global cognitive composite scores [41,45] as well as in specific cognitive tests such as memory (CVLT). Furthermore, this trajectory has also been observed in MMSE, memory composites, and PACC scores in SCD subjects [39,71,85] and in memory tasks among Aβ+ CU individuals [91]. In contrast, some studies have found no cross-sectional association between plasma NfL and MMSE scores [71], as well as memory and EF composite scores measured by ADNI-Memory and ADNI-EF [91], or with any cognitive scores across different domains [81,82] in CU individuals. Additionally, some longitudinal studies have not reported any association between baseline plasma NfL levels and a subsequent decline in cognitive scores, including those measured by the MMSE, ADASCog-11, and specific domain tasks such as episodic memory or SF [76,88,92]. This association was not found to be statistically significant for MMSE and PACC scores in individuals with CU Aβ+ individuals [44].

### 2.6. Inflammation: Blood-Based Glial Fibrillary Acid Protein (GFAP)

GFAP is a marker of astrogliosis and plays a critical role in maintaining cell structure as one of the cytoskeletal proteins within astrocytes [93]. Astrogliosis is a pathological process commonly associated with Aβ pathology in AD [94]. A recent study found that CSF GFAP levels are associated with Aβ [95], highlighting the potential role of neuroinflammation in amyloid plaque formation. Firstly, Oeckl et al. (2019) [96] discovered that serum GFAP is a valuable tool for distinguishing AD patients from the controls and those with frontotemporal dementia, and it could also serve as a CSF-independent marker. More recently, it has been reported that plasma GFAP levels are elevated in AD patients compared to CU individuals [77]. A recent meta-analysis has confirmed that astrocyte biomarkers are altered in AD, thus supporting their inclusion in clinical research on AD [97].

#### Association between Blood-Based GFAP and Cognition

A recent study comprising mixed cohorts showed that higher plasma GFAP was associated with lower language tests such as SF and BNT but also with lower cognitive scores such as attention/EF scores (LT, Design Fluency, SCWT, TMT and DS), visual memory scores (Benson Figure), and visuospatial scores (Benson Figure copy and Number Location of subtest of the Visual Object and Space Perception battery) [98]. Oeckl et al. (2019) [96] showed that elevated serum GFAP levels were also associated with lower MMSE scores but not with CDR-SOB scores. Furthermore, longitudinal data found that plasma GFAP levels were associated with greater declines in annual MMSE [30] and multi-domain composites scores [65].

In samples involving only CI subjects, a recent study with familial AD subjects showed a significant inverse cross-sectional association between plasma GFAP with MMSE, a cognitive composites score (composed by Logical Memory, Word List Learning, Digit Symbol and MMSE) and CDR-SOB scores. They also found that plasma GFAP was predictive of longitudinal declines in MMSE and CDR-SOB scores in mutation carriers [99]. Along this same line, another study, involving MCI with positive AD biomarkers, found that higher plasma GFAP was associated with annual ACE-R scores slopes [37]. However, a recent study conducted by Saloner et al. (2023) [67] showed weaker associations between GFAP and cognitive trajectories in MCI subjects.

Considering CU individuals, a longitudinal study discovered that baseline plasma GFAP significantly predicted a lower cognitive composite score in an ageing cohort [45]. Another recent cross-sectional study found that lower scores on both MMSE and PACC were correlated with higher plasma GFAP levels [69]. Otherwise, a study showed no longitudinal association with MMSE and PACC scores in Aβ+ CU individuals [44].

**Table 1 biomolecules-14-00093-t001:** Summary of studies examining the association between plasma amyloid levels and neuropsychological tests.

Author	Population	Study	Neuropsychological Test	Correlated
Xiao et al., 2021 [29]	Mixed sample	cross-sectional	MMSE; single-domain composites	↑ (Aβ42, Aβ42/Aβ40)
Sun et al., 2022 [30]	Mixed sample	cross-sectional	MMSE; Story Recall; CDT	↑ (Aβ42/Aβ40)
Sapkota et al., 2022 [31]	Mixed sample	longitudinal	Single-domain composites	↑↑ (Aβ42/Aβ40)
Tsai et al., 2019 [32]	Mixed sample	cross-sectional longitudinal	MMSE MMSE	No (Aβ42/Aβ40)
Pereira et al., 2021 [33]	Mixed sample	longitudinal	MMSE	No (Aβ42/Aβ40)
Chen et al., 2022 [34]	Mixed sample	longitudinal	MMSE, AdasCog13	No (Aβ42/Aβ40)
Chen et al., 2019 [35]	AD AD and MCI MCI	cross-sectional cross-sectional longitudinal	Single and multi-domain composites Multi-domain composites Single-domain composites	↑ (Aβ42) ↑ (Aβ42) ↑ (Aβ42/Aβ40)
Tsai et al., 2020 [36]	Amnestic MCI CU	longitudinal longitudinal	MMSE MMSE	↑ (Aβ42) No (Aβ42)
Chouliaras et al., 2022 [37]	MCI with positive AD biomarkers	cross-sectional and longitudinal	ACE-R	↑ (Aβ42/Aβ40)
Pascual-Lucas et al., 2023 [38]	CU	cross-sectional	The Face–Name Associative Memory Exam	↑ (Aβ42/Aβ40)
Cullen et al., 2021 [39]	CU	longitudinal	PACC	↑↑ (Aβ42/Aβ40)
Giudici et al., 2020 [40]	CU	longitudinal	Multi-domain composite	↑↑ (Aβ42/Aβ40)
Aschenbrenner al., 2022 [41]	CU	longitudinal cross-sectional	Multi-domain composite Multi-domain composite	↑↑ (Aβ42/Aβ40) No (Aβ42/Aβ40)
Hong et al., 2023 [42]	CU	longitudinal	Verbal Learning Test scores; TMT	↑↑ (Aβ42/Aβ40)
Simrén et al., 2021 [43]	CU	longitudinal	MMSE	No (Aβ42/Aβ40)
Ashton et al., 2022 [44]	CU	longitudinal	MMSE, PACC	No (Aβ42/Aβ40)
Saunders et al., 2023 [45]	CU	longitudinal	WAIS-III subtests	No (Aβ42/Aβ40)
Wang et al., 2022 [46]	CU	longitudinal	MMSE	No (Aβ42/Aβ40)

Cognitively unimpaired (CU); mild cognitive impairment (MCI); Alzheimer’s disease (AD); Mini-Mental State Examination (MMSE); Alzheimer’s Disease Assessment Scale-Cognitive Subscale (ADASCog); Auditory Verbal Learning Test (AVLT); Clock Drawing Test (CDT); Word List Learning (WLL); Trail Making Test (TMT); Modified Rey–Osterrieth Complex Figure Test (MROCFT); Addenbrooke’s Cognitive Examination Revised version (ACE-R); Preclinical Alzheimer’s Cognitive Composite (PACC); Wechsler Adult Intelligence Scale (WAIS). ↑↑ strong positive correlation defined by larger sample sizes or multivariate analyses. ↑ positive correlation defined by univariate analysis or in studies with smaller sample sizes.

**Table 2 biomolecules-14-00093-t002:** Summary of studies examining the association between different plasma Tau proteins and neuropsychological tests.

Author	Population	Study	Neuropsychological Test	Correlated
Tsai et al., 2019 [32]	Mixed sample MCI and AD	cross-sectional longitudinal	MMSE MMSE	↓ (p-tau181; t-tau) ↓ (t-tau); No (p-tau)
Karikari et al., 2020 [60]	Mixed sample	cross-sectional longitudinal	MMSE	↓↓ (p-tau181)
Xiao et al., 2021 [29]	Mixed sample	cross-sectional	MMSE, single-domain composites	↓ (p-tau181; t-tau)
Wang et al., 2021 [59]	Mixed sample MCI with positive AD biomarkers AD	cross-sectional longitudinal	MMSE; MoCA, ADNI single-domain composites ADNI-memory composite	↓ (p-tau181)
Sun et al., 2022 [30]	Mixed sample	cross-sectional	Story Recall, CDT	↓ (p-tau181; t-tau)
Weigand et al., 2023 [61]	Mixed sample	cross-sectional	AVLT	↓↓ (p-tau181)
Pereira et al., 2021 [33]	Mixed sample	longitudinal	MMSE	↓↓ (p-tau181, p-tau217)
Chen et al., 2022 [34]	Mixed sample	longitudinal	Adas-Cog13	↓↓ (p-tau181)
Smirnov et al., 2022 [62]	Mixed sample	longitudinal	DRS	↓↓ (p-tau181; p-tau231)
Tsai et al., 2020 [36]	Mixed sample	longitudinal	MMSE	No (p-tau181; t-tau)
Groot et al., 2023 [63]	Mixed sample	longitudinal	MMSE	↓ (p-tau217)
Pase et al., 2019 [49]	Mixed sample	cross-sectional	HVOT, Logical memory, Paired Associate Learning, Visual reproductions, TMT	↓↓ (t-tau)
Marks et al., 2021 [64]	Mixed sample	cross-sectional longitudinal	AVLT, Logical Memory (WMS-R) AVLT, Logical Memory, TMT, Digit Symbol (WAIS-R)	↓↓ (t-tau)
Sapkota et al., 2022 [31]	Mixed sample	longitudinal	Single-domain composites	↓↓ (t-tau)
Rajan et al., 2020 [65]	Mixed sample	longitudinal	Multi-domain composite	↓↓ (serum t-tau)
Smirén et al., 2021 [43]	MCI AD	longitudinal	MMSE MMSE	↓↓ (p-tau181)
Saloner et al., 2023 [67]	MCI CU	longitudinal longitudinal	single-domain composites CVLT	↓ (p-tau217)
Chouliaras et al., 2022 [37]	MCI and AD	longitudinal	ACE-R	↓ (p-tau181)
Moscoso et al., 2021 [66]	CU MCI and AD	longitudinal longitudinal	PACC ADASCog	↓↓ (p-tau181) ↓↓ (p-tau181)
Thomas et al., 2021 [68]	CU	longitudinal	PACC, CDR-SOB, FAQ	↓↓ (p-tau181)
Saunders et al., 2023 [45]	CU	longitudinal	WAIS-III subtests	↓↓ (p-tau181)
Ashton et al., 2022 [44]	CU	longitudinal	MMSE, PACC, RAVL	↓↓ (p-tau217)
Cullen et al., 2021 [39]	CU	longitudinal	PACC	↓↓ (p-tau217)
Mattsson-Carlgren et al., 2023 [70]	CU	longitudinal	MMSE, PACC	↓↓ (p-tau217)
Snellman et al., 2023 [69]	CU	cross-sectional	MMSE	No (p-tau181)
Baldacci et al., 2020 [71]	CU	cross-sectional longitudinal	MMSE, FCSRT MMSE	No (t-tau) ↓ (t-tau)
Rübsamen et al., 2021 [72]	CU	cross-sectional	Single-domain composite	No (t-tau)

Cognitively unimpaired (CU); mild cognitive impairment (MCI); Alzheimer’s disease (AD); Mini-Mental State Examination (MMSE); Montreal Cognitive Assessment (MoCA); Alzheimer’s Disease Neuroimaging Initiative (ADNI); Auditory Verbal Learning Test (AVLT); Clock Drawing Test (CDT); Trail Making Test (TMT); Addenbrooke’s Cognitive Examination Revised version (ACE-R); Preclinical Alzheimer’s Cognitive Composite (PACC); Clinical Dementia Rating Scale (CDR)- Sum Of Boxes (CDR-SOB), Functional Assessment Questionnaire (FAQ); Hooper Visual Organization Test (HVOT); Dementia Rating Scale (DRS); California Verbal Learning Test (CVLT); Alzheimer’s Disease Assessment Scale-Cognitive Subscale (ADASCog); Wechsler Adult Intelligence Scale (WAIS); Wechsler Memory Scale (WMS); Free and Cued Selective Reminding Test (FCSRT). ↓↓ strong negative correlation defined by larger sample sizes or multivariate analyses. ↓ negative correlation defined by univariate analysis or in studies with smaller sample sizes.

## 3. Discussion

### 3.1. Blood-Based Aβ

Despite several confirmatory significant associations reported in some studies, the relationship between blood-based Aβ, particularly the Aβ_42_/Aβ_40_ ratio, and cognitive performance exhibits contradictory findings. Mixed cohorts (CU and impaired individuals) often show a positive correlation between blood-based Aβ_42_ and/or the Aβ_42_/Aβ_40_ ratio and cognition, as measured by the MMSE or other specific cognitive domains such as memory, attention/EF, and visuospatial measures. Similarly, studies with CI cohorts, such as the one by Chen et al. (2019) [35], reveal positive correlations with various cognitive composites, while others do not consistently observe such findings. In studies involving only CI individuals, the relationship between plasma Aβ levels and cognitive decline is less clear, possibly due to a plateau in biomarker sensitivity or overshadowing by other pathological processes. Among CU individuals, cross-sectional studies do not find associations between blood-based Aβ and cognitive performance, whereas some longitudinal studies report associations between decline in cognitive performance associated with a lower blood-based Aβ_42_/Aβ_40_ ratio. This inconsistency may stem from variations in study populations and disease stages. Furthermore, the variability in the sensitivity and specificity of the cognitive tests used in these studies, such as MMSE scores, contributes to the mixed findings, as these tests may not be sensitive enough to detect subtle cognitive changes associated with variations in plasma Aβ levels. In addition, as mentioned earlier, blood-based Aβ measures are not as robust as CSF Aβ or other blood-based AD biomarkers due to technical reasons. These differences may contribute to the contradictory findings reported in some research works. Moreover, some of these studies lack in vivo confirmation of AD pathophysiology using well-established CSF core AD biomarkers or brain Amyloid PET scans.

### 3.2. Blood-Based Total Tau and Phosphorylated Tau

Blood-based Tau biomarkers, including p-Tau181, p-Tau217, p-Tau231, and t-Tau, consistently demonstrate a negative correlation with cognitive performance. Furthermore, studies within our review suggest that blood-based p-Tau measures show stronger associations with cognition than other blood-based biomarkers [29,34,43,44,45,66,67,70]. Thus, these findings suggest that plasma p-Tau measures are a sensitive marker in AD-related cognitive impairment. Elevated plasma p-Tau181 levels are associated with poorer performances in both global and domain-specific cognitive tests among mixed samples of CU and impaired individuals. However, some studies, such as Tsai et al. (2020) [36], do not observe this association. As is the case in previous studies with blood-based Aβ, this variability may be attributed to differences in study populations, stages of disease progression, and sensitivities of cognitive test. In addition to this, as mentioned before, some of these studies lack in vivo confirmation of AD pathophysiology using well-established AD biomarkers. In CI groups, blood-based Tau biomarkers negatively correlate with performance in various cognitive domains. The relationship between blood-based Tau biomarkers and cognitive outcomes seems to be more pronounced in clinically diagnosed populations, suggesting a direct linkage to disease pathology. Regarding CU individuals, the cross-sectional studies included in this review do not report associations between blood-based p-Tau nor t-Tau and cognitive performance. However, some longitudinal studies have suggested that plasma p-Tau and measures can predict cognitive decline during the earliest AD stages, suggesting that a lower increase in p-Tau concentrations points to preclinical stages of AD.

### 3.3. Blood-based NfL

Blood-based NfL exhibits a negative correlation with cognitive performance, as evidenced in both cross-sectional and longitudinal studies. This association is noted in global cognitive scores and specific cognitive domains, including attention/EF, memory, and language. However, some studies, like Mielke et al. (2019) [84] and Pereira et al. (2021) [33], failed to find a significant association, underscoring once the heterogeneity in study populations, the characteristics of cognitive tests used, and a low specificity of this biomarker of AD. In CI populations, higher blood-based NfL levels correlate with a poorer performance across various cognitive domains. However, in CU individuals, the predictive value of blood-based NfL for cognitive decline appears to be less consistent. These observed differences may be attributed to the fact that blood-based NfL, serving as a marker of neurodegeneration, could be indicative of clinical progression and advanced symptoms of the disease.

### 3.4. Blood-Based GFAP

Recent studies indicate that elevated blood-based GFAP levels are associated with poorer cognitive function in both global cognition and specific cognitive measures. Some discrepancies in the results can be attributed to the stage of disease progression and the sensitivity of cognitive assessments. Among CU individuals, correlations between higher blood-based GFAP levels and lower cognitive scores have been observed, although not consistently across studies. For instance, Saloner et al. (2023) [67] reported weak associations in MCI subjects, possibly due to the biological heterogeneity of a clinical cohort. For CU individuals, studies also find an inverse relation, suggesting that blood-based GFAP could also be considered to be an early AD marker.

### 3.5. General Discussion

This descriptive review summarizes the main findings from the recent literature and underscores the complexity and heterogeneity in the relationships between blood-based biomarkers and cognitive outcomes in AD. As previously outlined, these disparities may stem from the heterogeneity in methodology and samples across the included studies, the intrinsic characteristics of the biomarker, technical variations in its measurement techniques, and its correlation with the disease stage. Additionally, they may also be influenced by the variability in the properties of the cognitive tests employed. In this descriptive review, no single blood-based biomarker has shown a predominant association with a specific domain. One reason for this could be that many studies use a multi-domain composite score to measure cognitive performance, which prevents a more specific analysis. Otherwise, blood-based p-Tau measures, exhibit stronger associations with cognition than other biomarkers, highlighting their potential in clinical and research settings. An emerging question in AD research is whether blood-based biomarkers could potentially replace CSF measures in the future. Currently, CSF biomarkers are considered the standard of truth due to their direct reflection of brain pathology. However, the invasive nature of CSF collection poses limitations. On the other hand, blood-based biomarkers offer a less invasive alternative. While blood-based biomarkers are showing promising correlations with currently used AD biomarkers (amyloid brain PET scan and CSF core AD biomarkers), there are still challenges regarding sensitivity and specificity compared to CSF measures, especially for Aβ measures. Notably, advancements in technologies like SIMOA have shown potential in overcoming these challenges. SIMOA’s high-sensitivity detection capabilities make it a promising tool for accurately measuring neurological biomarkers in blood, which may facilitate early detection and monitoring of AD progression. Future research should focus on refining these blood biomarkers’ analytical performance and validating their clinical utility in large-scale, diverse cohorts. The potential replacement of CSF biomarkers by blood-based methods depends not only on technological advancements but also on comprehensive validation studies that ensure their efficacy at various stages of the disease. Moreover, there is a need to explore their potential utility as prognostic markers and indicators of a possible response to potential disease-modifying treatments, aligning with neuropsychological assessment. This will enable a more personalized approach to diagnosis and treatment, improving patient outcomes. As the field evolves, it is imperative to continuously evaluate the roles of both blood-based and CSF diagnostics in AD, aiming for the most effective, patient-friendly, and accurate diagnostic methods.

### 3.6. Limitations

The search for this review was conducted exclusively using PubMed, which may have resulted in the omission of significant studies published in databases not indexed there. While the review incorporates all the relevant studies identified from the reference lists of initially retrieved articles, it is possible that some pertinent studies may have been overlooked. It should also be noted that this is a descriptive review, not a systematic review or meta-analysis; therefore, the conclusions drawn may lack the statistical power that could be achieved with a more rigorous methodology. These limitations should be considered when interpreting the findings of this review.

## 4. Conclusions

In conclusion, this review represents a preliminary approach to the association between blood-based biomarkers of AD and cognitive measures. While the potential for blood-based biomarkers of AD is promising, further extensive clinical validation is needed for routinary clinical application. The ongoing development in this area represents an exciting frontier in AD research, with the prospect of transforming diagnostic and treatment paradigms.

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
