# Peer review of "Blood Biomarkers of Alzheimer’s Disease and Cognition: A Literature Review"

_biomolecules, 2024, doi:10.3390/biom14010093_

Round 1

Reviewer 1 Report

Comments and Suggestions for Authors

This is a good start to a review on blood biomarkers for Alzheimer's disease.  The authors compile and repeat what is published in the literature, but do not add any insight into trying to compile the results. The authors do a reasonable job of reporting what has been published in literature on abeta, tau, neurofilament light and glial fibrillary Acid protein biomarkers. However the different publications used different methods of correlating the markers with different cognitive outcomes and stage of disease. The authors merely report these studies instead of trying to compile trends that are common or different between studies.  There are no recommendations on what looks to be the most promising biomarkers for different applications.  Therefore this is a good start to a review, but additional insight needs to be included to make the review helpful and to contribute to the field.  

Comments on the Quality of English Language

English needs substantial review to make the article easier to read and grammatically correct. 

Author Response

Thank you for your insightful feedback on our review regarding blood biomarkers for Alzheimer's disease. We have considered your comments and substantially revised the discussion section in our manuscript. In this revised version, we have focused on a more analytical approach, identifying common trends and discrepancies across studies, and providing deeper insights into the implications of these findings. Our revised discussion now includes detailed sections for each biomarker, addressing the complexities and variability in their relationships with cognitive outcomes. Specifically, we have included comprehensive discussions on Plasma Amyloid-β, Plasma Total Tau and Phosphorylated Tau, Plasma Neurofilament Light (NfL), and Plasma Glial Fibrillary Acidic Protein (GFAP).

The revised discussion section is as follows (see line 325):

“3. Discussion

Blood-based Aβ:

Despite several confirmatory significant associations reported in some studies, the relationship between blood-based Aβ, particularly the Aβ42/Aβ40 ratio, and cognitive performance exhibits contradictory findings. Mixed cohorts (CU and impaired individuals) often show a positive correlation between blood-based Aβ42 and/or the Aβ42/Aβ40 ratio and cognition, as measured by the MMSE or other specific cognitive domains such as memory, attention/EF, and visuospatial measures. Similarly, studies with CI cohorts, such as the one by Chen et al. (2019) [35], reveal positive correlations with various cognitive composites, while others do not consistently observe such findings. In studies involving only CI individuals, the relationship between plasma Aβ levels and cognitive decline is less clear, possibly due to a plateau in biomarker sensitivity or overshadowing by other pathological processes. Among CU individuals, cross-sectional studies do not find associations between blood-based Aβ and cognitive performance, whereas some longitudinal studies report associations between decline in cognitive performance associated with lower blood-based Aβ42/Aβ40 ratio. This inconsistency may stem from variations in study populations and disease stages. Furthermore, the variability in the sensitivity and specificity of cognitive tests used in these studies, such as MMSE scores, contributes to the mixed findings, as these tests may not be sensitive enough to detect subtle cognitive changes associated with variations in plasma Aβ level. In addition, as mentioned earlier, blood-based Aβ measures are not as robust as CSF Aβ or other blood-based AD biomarkers due to technical reasons. These differences may contribute to the contradictory findings reported in some research. Moreover, some of these studies lack in vivo confirmation of AD pathophysiology using well-established CSF core AD biomarkers or brain Amyloid PET scan.

Blood-based Total Tau and Phosphorylated Tau:

Blood-based tau biomarkers, including p-Tau181, p-Tau217, p-Tau231, and t-Tau, consistently demonstrate a negative correlation with cognitive performance. Furthermore, studies within our review suggest that blood-based p-Tau measures show stronger associations with cognition than other blood-based biomarkers [29][34][43][44][45][66] [67][70]. Thus, these findings suggest that plasma p-Tau measures are a sensitive marker in AD-related cognitive impairment. Elevated plasma p-Tau181 levels are associated with poorer performance in both global and domain-specific cognitive tests among mixed samples of CU and impaired individuals. However, some studies, such as Tsai et al. (2020) [36], do not observe this association. As is the case in previous studies with blood-based Aβ, this variability may be attributed to differences in study populations, stages of disease progression, and sensitivities of cognitive test. In addition to this, as mentioned before, some of these studies lack in vivo confirmation of AD pathophysiology using well stablished AD biomarkers. In CI groups, blood-based tau biomarkers negatively correlate with performance in various cognitive domains. The relationship between blood-based tau biomarkers and cognitive outcomes seems to be more pronounced in clinically diagnosed populations, suggesting a direct linkage to disease pathology. Regarding CU individuals, cross-sectional studies included in this review do not report association between blood-based p-Tau nor t-Tau and cognitive performance. However, some longitudinal studies have suggested that plasma p-Tau and measures can predict cognitive decline during earliest AD stages, suggesting that lower increase in p-Tau concentrations points to preclinical stages of AD.

Blood-based NfL:

Blood-based NfL exhibits a negative correlation with cognitive performance, as evidenced in both cross-sectional and longitudinal studies. This association is noted in global cognitive scores and specific cognitive domains, including attention/EF, memory, and language. However, some studies, like Mielke et al. (2019) [84], and Pereira et al. (2021) [33], failed to find a significant association, underscoring once the heterogeneity in study populations, characteristics of cognitive tests used and a low specificity of this biomarker for AD. In CI populations, higher blood-based NfL levels correlate with poorer performance across various cognitive domains. However, in CU individuals, the predictive value of blood-based NfL for cognitive decline appears less consistent. These observed differences may be attributed to the fact that blood-based NfL, serving as a marker of neurodegeneration, could be indicative of clinical progression and advanced symptoms of the disease.

Blood-based GFAP:

Recent studies indicate that elevated blood-based GFAP levels are associated with poorer cognitive function in both global cognition and specific cognitive measures. Some discrepancies in the results can be attributed to the stage of disease progression and the sensitivity of cognitive assessments. Among CU individuals, correlations between higher blood-based GFAP levels and lower cognitive scores have been observed, although not consistently across studies. For instance, Saloner et al. (2023) [67] reported weak associations in MCI subjects, possibly due to the biological heterogeneity of a clinical cohort. For CU individuals, studies also find an inverse relation, suggesting that blood-based GFAP could also be considered as an early AD marker.

General Discussion:

This descriptive review summarizes the main findings form recent literature and underscores the complexity and heterogeneity in the relationships between blood-based biomarkers and cognitive outcomes in AD. As previously outlined, these disparities may stem from the heterogeneity in methodology and samples across the included studies, the intrinsic characteristics of the biomarker, technical variations in its measurement techniques, and its correlation with the disease stage. Additionally, they may also be influenced by the variability in the properties of the cognitive tests employed. In this descriptive review, no single blood-based biomarker has shown a predominant association with a specific domain. One reason for this could be that many studies use a multi-domain composite score to measure cognitive performance, which prevents a more specific analysis. Otherwise, blood-based p-tau measures, however, exhibit stronger associations with cognition than other biomarkers, highlighting their potential in clinical and research settings. An emerging question in AD research is whether blood-based biomarkers could potentially replace CSF measures in the future. Currently, CSF biomarkers are considered the standard of truth due to their direct reflection of brain pathology. However, the invasive nature of CSF collection poses limitations. On the other hand, blood-based biomarkers offer a less invasive alternative. While blood-based biomarkers are showing promising correlations with currently used AD biomarkers (amyloid brain PET scan and CSF core AD biomarkers) there are still challenges regarding sensitivity and specificity compared to CSF measures, especially for Aβ measures. Notably, advancements in technologies like SIMOA have shown potential in overcoming these challenges. SIMOA's high-sensitivity detection capabilities make it a promising tool for accurately measuring neurological biomarkers in blood, which may facilitate early detection and monitoring of AD progression. Future research should focus on refining these blood biomarkers' analytical performance and validating their clinical utility in large-scale, diverse cohorts. The potential replacement of CSF biomarkers by blood-based methods depends not only on the technological advancements but also on comprehensive validation studies that ensure their efficacy in various stages of the disease. Moreover, there is a need to explore their potential utility as prognostic markers and indicators of possible response to potential disease-modifying treatments, aligning with neuropsychological assessment. This will enable a more personalized approach to diagnosis and treatment, improving patient outcomes. As the field evolves, it is imperative to continuously evaluate the roles of both blood-based and CSF diagnostics in AD, aiming for the most effective, patient-friendly, and accurate diagnostic methods.”

We believe these modifications provide a more thorough and insightful analysis, thereby enhancing the contribution of our review to the field. We hope that this revised version meets your expectations and addresses the concerns you have raised.

Reviewer 2 Report

Comments and Suggestions for Authors

Garcia-Escobar et al. present a review of the recent literature pertaining to the association between plasma biomarkers and cognitive measures related to AD. This is an exhaustive review that could be useful for researchers who are interested in a snapshot of current literature. 

However, there are some issues that need to be addressed. 

Materials and methods: 

Did the search include specific cognitive measures? Were specific cognitive tests or batteries selected? What were the exclusion criteria for the studies (populations, tests, etc.)? Were there any exclusion criteria for the biomarkers or were all included independently of the method and platform used?  

For the A section, on line 85, the authors talk about Abeta pathology, but they do not mention how was it measured. Is it PET measures or postmortem immunohistochemical evaluation? 

General comments about the body of the review. At times it reads more like a list of studies stating the results from each of them than a cohesive review. It would benefit from grouping studies that measure the same cognitive domain together. This would potentially solve the issue that sometimes it is not clear how different tests relate to each other, especially considering that some of the readers will not be familiar with all the tests mentioned in the text. 

In the case of studies that have conflicting results on the same biomarker and same cognitive domain. How do the authors explain those discrepancies? What are the differences between studies? Population, sex distribution, age distribution, education levels, APOE genotypes? 

On line 323 the authors write "potentially indicating a link between Abeta and Tau pathology" about GFAP and referencing a review, however, the review does not mention either of those things. I found it startling because in the previous sentences, the authors were talking about GFAP and amyloid deposition, which would link neuroinflammation with amyloid plaques, but there is no previous mention of tau. 

Lines 371 to 376 are out of place in the discussion. Those should have been included in methods for the reader to understand the different tests that are mentioned.

For the tables, it would be good to have the direction of the correlation, not only whether it was correlated or not.

Comments on the Quality of English Language

This paper needs extensive language editing. Some of the sentences are extremely verbose and there are instances of the wrong verb being used. There is an excessive use of prepositions that make the manuscript feel like a listing rather than a literature review. I feel like speakers of other languages will have a hard time reading this review. 

Author Response

Garcia-Escobar et al. present a review of the recent literature pertaining to the association between plasma biomarkers and cognitive measures related to AD. This is an exhaustive review that could be useful for researchers who are interested in a snapshot of current literature. 

Thank you for your thoughtful review and constructive suggestions regarding our manuscript. We appreciate your interest in our work and agree that clarifying our methods will enhance the quality and comprehensibility of our review.

However, there are some issues that need to be addressed. 

Materials and methods: 

Did the search include specific cognitive measures? Were specific cognitive tests or batteries selected? What were the exclusion criteria for the studies (populations, tests, etc.)? Were there any exclusion criteria for the biomarkers or were all included independently of the method and platform used?  

Thank you for your constructive feedback on our manuscript. We have thoroughly revised our "Materials and Methods" section to address your concerns. Specifically, we have detailed our search strategy, inclusion and exclusion criteria, and the methodological approach employed in our literature review. Additionally, we have incorporated a peer review process into our methodology. Three senior researchers to verify their scientific quality and relevance to our review reviewed all preselected articles from the initial search. This step was taken to ensure that only the most pertinent and robust studies were included in our analysis. We believe these revisions provide a clearer and more comprehensive understanding of our methodological approach, addressing the issues you raised. The updated "Materials and Methods" section now accurately reflects the thorough and meticulous process we employed in conducting this review (see line 54):

“Search Strategy and Selection Criteria:

We conducted a comprehensive literature search on PubMed using the strategy: 'Plasma biomarkers’ AND ‘Alzheimer's disease’ AND ’Cognitive change.’ This search specifically targeted articles published within the last five years (2018-2023).

Inclusion and Exclusion Criteria:

Our inclusion criteria focused on studies that investigated blood-based biomarkers for AD, related neurodegeneration and glial activation (Aβ, p-Tau, t-Tau, NfL, and GFAP) in human subjects. We selected studies that utilized recognized cognitive assessments and presented findings from both cross-sectional and longitudinal study designs. The study populations were diverse, including samples with both cognitively impaired (CI) and cognitively unimpaired (CU) subjects (mixed population), as well as those exclusively comprising either CI or unimpaired individuals. We excluded studies not directly addressing AD, lacking clear cognitive outcome measures, or based on non-human subjects or in vitro models.

Peer Review Process

All preselected articles identified through our search criteria were further reviewed by three senior researchers (GGE, AO, APP). This rigorous review process was implemented to verify the quality and relevance of each study, ensuring the inclusion of only the most pertinent and scientifically robust research in our review.

Biomarkers measurements

The recent development of advanced technologies has significantly enhanced the detection capabilities for plasma proteins. In the articles included in this review, various techniques and platforms have been employed to determine disease-related proteins with high sensitivity and precision, even at low concentrations. Among these platforms is SIMOA (Single Molecule Array, Quanterix Corporation), a digital immunoassay technique that enables the detection of biomarkers at extremely low concentrations in blood samples. This technology is particularly significant for AD research, as it allows for the accurate measurement of neurological biomarkers present in minute quantities, thereby facilitating early detection and monitoring of disease progression. Other notable platforms include MSD (Meso Scale Discovery, Meso Scale Diagnostics, LLC), LC-MS/MS (Liquid Chromatography-Mass Spectrometry), ELISA (Enzyme-Linked Immunosorbent Assay), AlphaLISA (Amplified Luminescent Proximity Homogeneous Assay, PerkinElmer), and Olink Proteomics (Olink Proteomics AB). The detailed methodologies for each platform are outlined in the cited articles.

Data Extraction and Synthesis:

Data were extracted from the selected articles, focusing on the relationship between blood-based biomarkers and cognitive measures in AD. We emphasized studies using established neuropsychological tests across cognitive domains. Global cognition was primarily assessed using the Mini-Mental State Examination (MMSE) [11] and the Montreal Cognitive Assessment [12]. The memory domain was evaluated through tasks like the Auditory Verbal Learning Test (AVLT) [13] and the Free and Cued Selective Reminding Test [14], while visuospatial abilities were often measured using tests like the Clock Drawing Test (CDT) [15]; historical review] Language abilities were examined with tests such as semantic fluency (SF) or the Boston Naming Test [16], and attention/ EF domains were assessed using tests like verbal span,  the Trail Making Test (TMT) [17]  and Stroop Color and Word Test [18].  This inclusion helps underline the ongoing validity of these neuropsychological tests in both routine clinical practice and research settings.

Presentation of Findings:

The findings are synthesized descriptively, highlighting the correlations between blood-based biomarkers and cognitive measures in the AD continuum. Summary tables (table 1 and 2) are included to provide a clear and concise overview of the observed associations of the main blood-based AD biomarkers: plasma Aβ and p-Tau and t-Tau measures. To enhance the clarity and accuracy of our review, we have implemented a systematic approach for assigning directional arrows in the correlation tables, which reflect the strength and nature of the correlations reported in the studies. This evaluation process was meticulously carried out by two senior neurologists and one neuropsychologist, ensuring a thorough and expert analysis of each study's findings. A single upward arrow (↑) indicates a positive correlation identified through univariate analysis or in studies with smaller sample sizes. Two upward arrows (↑↑) denote a stronger positive correlation, typically observed in studies with larger sample sizes or multivariate analyses. For negative correlations, where higher biomarker levels correspond to poorer cognitive performance or faster cognitive decline, a single downward arrow (↓) is used for univariate analyses or smaller studies, and two downward arrows (↓↓) are employed for stronger negative correlations identified in larger sample sizes or multivariate analyses.

  1. Blood-based Aβ

The accumulation of Aβ peptides, associated with AD pathophysiology, becomes detectable up to 20-30 years before the onset of clinical dementia [19]. Various studies have evaluated the accuracy of blood-based Aβ biomarkers in detecting Aβ pathology, using brain amyloid PET scans and/or CSF Aβ biomarkers as the gold standard  [20, 21, 22]. While blood-based measures of Aβ offer valuable insights, it is important to note that their reliability tends to be poorer compared to Aβ measures in CSF. While blood-based Aβ measures provide valuable insights, it is important to note that their reliability tends to be lower compared to Aβ measures in CSF. Additionally, studies have found lower plasma Aβ levels in individuals with AD-type dementia compared to CU individuals and other diagnostic groups [23], although these blood-based measures tend to show less variation between groups than those in CSF [24]. Furthermore, plasma Aβ levels have demonstrated the ability to differentiate AD-type dementia from Mild Cognitive Impairment (MCI) patients [25] with MCI classically defined as cognitive impairment without meeting dementia criteria [26]. On the other hand, some studies have reported that the Aβ42/40 ratio in plasma exhibits a better correlation with Aβ pathology than individual Aβ42 and Aβ40 measures alone [27, 28].”

For the A section, on line 85, the authors talk about Abeta pathology, but they do not mention how was it measured. Is it PET measures or postmortem immunohistochemical evaluation? 

Thank you for your comment regarding the clarity of our discussion on Aβ pathology measurement methods in section A, line 85. We understand the importance of specifying the methodologies used in assessing Aβ pathology. The focus of our discussion in this section is on plasma biomarkers. To clarify this, we have revised the beginning of the paragraph to explicitly state the method of measurement. The revised paragraph now reads (line 103):

A. Blood-based Aβ

The accumulation of Aβ peptides, associated with AD pathophysiology, becomes detectable up to 20-30 years before the onset of clinical dementia [19]. Various studies have evaluated the accuracy of blood-based Aβ biomarkers in detecting Aβ pathology, using brain amyloid PET scans and/or CSF Aβ biomarkers as the gold standard  [20, 21, 22]. While blood-based measures of Aβ offer valuable insights, it is important to note that their reliability tends to be poorer compared to Aβ measures in CSF. While blood-based Aβ measures provide valuable insights, it is important to note that their reliability tends to be lower compared to Aβ measures in CSF. Additionally, studies have found lower plasma Aβ levels in individuals with AD-type dementia compared to CU individuals and other diagnostic groups [23], although these blood-based measures tend to show less variation between groups than those in CSF [24]. Furthermore, plasma Aβ levels have demonstrated the ability to differentiate AD-type dementia from Mild Cognitive Impairment (MCI) patients [25] with MCI classically defined as cognitive impairment without meeting dementia criteria [26]. On the other hand, some studies have reported that the Aβ42/40 ratio in plasma exhibits a better correlation with Aβ pathology than individual Aβ42 and Aβ40 measures alone [27, 28].”

 General comments about the body of the review. At times it reads more like a list of studies stating the results from each of them than a cohesive review. It would benefit from grouping studies that measure the same cognitive domain together. This would potentially solve the issue that sometimes it is not clear how different tests relate to each other, especially considering that some of the readers will not be familiar with all the tests mentioned in the text. 

In the case of studies that have conflicting results on the same biomarker and same cognitive domain. How do the authors explain those discrepancies? What are the differences between studies? Population, sex distribution, age distribution, education levels, APOE genotypes? 

Thank you for your valuable feedback on the structure and content of our review. Your suggestions have led us to significantly revise our manuscript to ensure a more cohesive and insightful presentation of the research in this field. In response to your concerns, we have meticulously restructured the discussion section of our manuscript. We now categorize studies based on the cognitive domains they measure, which provides a clearer understanding of how different tests interrelate and offers a more coherent overview of the field. This restructuring helps in delineating the connections between various cognitive assessments and their implications in Alzheimer's disease research. Additionally, we have addressed the issue of conflicting results observed in studies examining the same biomarkers within the same cognitive domains. The revised discussion delves into possible reasons for these discrepancies, such as variations in study populations, sex and age distribution, education levels, and APOE genotypes. Through this detailed examination, we aim to provide a more comprehensive analysis of the results and contribute to a better understanding of the complex nature of Alzheimer's disease biomarkers. Below is the revised discussion that reflects these changes (see line 325) and conclusions (see line 422):

3. Discussion

Blood-based Aβ:

Despite several confirmatory significant associations reported in some studies, the relationship between blood-based Aβ, particularly the Aβ42/Aβ40 ratio, and cognitive performance exhibits contradictory findings. Mixed cohorts (CU and impaired individuals) often show a positive correlation between blood-based Aβ42 and/or the Aβ42/Aβ40 ratio and cognition, as measured by the MMSE or other specific cognitive domains such as memory, attention/EF, and visuospatial measures. Similarly, studies with CI cohorts, such as the one by Chen et al. (2019) [35], reveal positive correlations with various cognitive composites, while others do not consistently observe such findings. In studies involving only CI individuals, the relationship between plasma Aβ levels and cognitive decline is less clear, possibly due to a plateau in biomarker sensitivity or overshadowing by other pathological processes. Among CU individuals, cross-sectional studies do not find associations between blood-based Aβ and cognitive performance, whereas some longitudinal studies report associations between decline in cognitive performance associated with lower blood-based Aβ42/Aβ40 ratio. This inconsistency may stem from variations in study populations and disease stages. Furthermore, the variability in the sensitivity and specificity of cognitive tests used in these studies, such as MMSE scores, contributes to the mixed findings, as these tests may not be sensitive enough to detect subtle cognitive changes associated with variations in plasma Aβ level. In addition, as mentioned earlier, blood-based Aβ measures are not as robust as CSF Aβ or other blood-based AD biomarkers due to technical reasons. These differences may contribute to the contradictory findings reported in some research. Moreover, some of these studies lack in vivo confirmation of AD pathophysiology using well-established CSF core AD biomarkers or brain Amyloid PET scan.

Blood-based Total Tau and Phosphorylated Tau:

Blood-based tau biomarkers, including p-Tau181, p-Tau217, p-Tau231, and t-Tau, consistently demonstrate a negative correlation with cognitive performance. Furthermore, studies within our review suggest that blood-based p-Tau measures show stronger associations with cognition than other blood-based biomarkers [29][34][43][44][45][66] [67][70]. Thus, these findings suggest that plasma p-Tau measures are a sensitive marker in AD-related cognitive impairment. Elevated plasma p-Tau181 levels are associated with poorer performance in both global and domain-specific cognitive tests among mixed samples of CU and impaired individuals. However, some studies, such as Tsai et al. (2020) [36], do not observe this association. As is the case in previous studies with blood-based Aβ, this variability may be attributed to differences in study populations, stages of disease progression, and sensitivities of cognitive test. In addition to this, as mentioned before, some of these studies lack in vivo confirmation of AD pathophysiology using well stablished AD biomarkers. In CI groups, blood-based tau biomarkers negatively correlate with performance in various cognitive domains. The relationship between blood-based tau biomarkers and cognitive outcomes seems to be more pronounced in clinically diagnosed populations, suggesting a direct linkage to disease pathology. Regarding CU individuals, cross-sectional studies included in this review do not report association between blood-based p-Tau nor t-Tau and cognitive performance. However, some longitudinal studies have suggested that plasma p-Tau and measures can predict cognitive decline during earliest AD stages, suggesting that lower increase in p-Tau concentrations points to preclinical stages of AD.

Blood-based NfL:

Blood-based NfL exhibits a negative correlation with cognitive performance, as evidenced in both cross-sectional and longitudinal studies. This association is noted in global cognitive scores and specific cognitive domains, including attention/EF, memory, and language. However, some studies, like Mielke et al. (2019) [84], and Pereira et al. (2021) [33], failed to find a significant association, underscoring once the heterogeneity in study populations, characteristics of cognitive tests used and a low specificity of this biomarker for AD. In CI populations, higher blood-based NfL levels correlate with poorer performance across various cognitive domains. However, in CU individuals, the predictive value of blood-based NfL for cognitive decline appears less consistent. These observed differences may be attributed to the fact that blood-based NfL, serving as a marker of neurodegeneration, could be indicative of clinical progression and advanced symptoms of the disease.

Blood-based GFAP:

Recent studies indicate that elevated blood-based GFAP levels are associated with poorer cognitive function in both global cognition and specific cognitive measures. Some discrepancies in the results can be attributed to the stage of disease progression and the sensitivity of cognitive assessments. Among CU individuals, correlations between higher blood-based GFAP levels and lower cognitive scores have been observed, although not consistently across studies. For instance, Saloner et al. (2023) [67] reported weak associations in MCI subjects, possibly due to the biological heterogeneity of a clinical cohort. For CU individuals, studies also find an inverse relation, suggesting that blood-based GFAP could also be considered as an early AD marker.

General Discussion:

This descriptive review summarizes the main findings form recent literature and underscores the complexity and heterogeneity in the relationships between blood-based biomarkers and cognitive outcomes in AD. As previously outlined, these disparities may stem from the heterogeneity in methodology and samples across the included studies, the intrinsic characteristics of the biomarker, technical variations in its measurement techniques, and its correlation with the disease stage. Additionally, they may also be influenced by the variability in the properties of the cognitive tests employed. In this descriptive review, no single blood-based biomarker has shown a predominant association with a specific domain. One reason for this could be that many studies use a multi-domain composite score to measure cognitive performance, which prevents a more specific analysis. Otherwise, blood-based p-tau measures, however, exhibit stronger associations with cognition than other biomarkers, highlighting their potential in clinical and research settings. An emerging question in AD research is whether blood-based biomarkers could potentially replace CSF measures in the future. Currently, CSF biomarkers are considered the standard of truth due to their direct reflection of brain pathology. However, the invasive nature of CSF collection poses limitations. On the other hand, blood-based biomarkers offer a less invasive alternative. While blood-based biomarkers are showing promising correlations with currently used AD biomarkers (amyloid brain PET scan and CSF core AD biomarkers) there are still challenges regarding sensitivity and specificity compared to CSF measures, especially for Aβ measures. Notably, advancements in technologies like SIMOA have shown potential in overcoming these challenges. SIMOA's high-sensitivity detection capabilities make it a promising tool for accurately measuring neurological biomarkers in blood, which may facilitate early detection and monitoring of AD progression. Future research should focus on refining these blood biomarkers' analytical performance and validating their clinical utility in large-scale, diverse cohorts. The potential replacement of CSF biomarkers by blood-based methods depends not only on the technological advancements but also on comprehensive validation studies that ensure their efficacy in various stages of the disease. Moreover, there is a need to explore their potential utility as prognostic markers and indicators of possible response to potential disease-modifying treatments, aligning with neuropsychological assessment. This will enable a more personalized approach to diagnosis and treatment, improving patient outcomes. As the field evolves, it is imperative to continuously evaluate the roles of both blood-based and CSF diagnostics in AD, aiming for the most effective, patient-friendly, and accurate diagnostic methods.”

“In conclusion, this review represents a preliminary approach to the associations blood-based biomarkers for AD and cognitive measures While the potential for blood-based biomarkers for AD is promising, further extensive clinical validation is needed for routinary clinical application. The ongoing development in this area represents an exciting frontier in AD research, with the prospect of transforming diagnostic and treatment paradigms.”

Additionally, in response to your feedback, we have made modifications to our tables. These changes are aimed at succinctly representing the methodological differences among the studies, particularly focusing on whether the studies employed multivariate adjustments and the relative sizes of their sample populations. To achieve this, we have added arrows to the tables as follows: A single arrow (either upward or downward) is used to denote studies that conducted univariate analysis or had smaller sample sizes. Two arrows (either upward or downward) represent studies that utilized multivariate analysis methods or had larger sample sizes. This methodological representation using arrows allows for a quick and clear understanding of the study design and analytical approach, thereby providing a concise overview of their differences as you suggested. The specifics of this approach are detailed in our revised Materials and Methods section, which outlines the criteria for the assignment of one or two arrows (see line 90).

We believe these enhancements will greatly aid in the comparison and comprehension of the diverse studies included in our review.

On line 323 the authors write "potentially indicating a link between Abeta and Tau pathology" about GFAP and referencing a review, however, the review does not mention either of those things. I found it startling because in the previous sentences, the authors were talking about GFAP and amyloid deposition, which would link neuroinflammation with amyloid plaques, but there is no previous mention of tau. 

Thank you for pointing out the discrepancy in our manuscript regarding the reference to a potential link between Aβ and Tau pathology in relation to GFAP. Upon re-evaluating this section, we agree with your observation that the mentioned review (reference 10) does not explicitly discuss this link. Consequently, we have revised line 323 to reflect the current understanding of GFAP’s relationship more accurately with amyloid deposition, as follows (see line 300): “A recent study found that CSF GFAP levels are associated with Aβ [95], highlighting the potential role of neuroinflammation in amyloid plaque formation.” We have removed the incorrect assertion about the link between Aβ and Tau pathology and omitted reference 10 from this sentence.

Lines 371 to 376 are out of place in the discussion. Those should have been included in methods for the reader to understand the different tests that are mentioned.

Thank you for your valuable feedback. We agree with your suggestion that the lines 371 to 376, which describe the neuropsychological tests used in the studies, would be more appropriately placed in the Materials and Methods section. We have revised our manuscript accordingly to enhance clarity and readability. This change ensures that readers have a clear understanding of the cognitive measures employed in the studies we reviewed, right from the outset.

For the tables, it would be good to have the direction of the correlation, not only whether it was correlated or not.

Thank you for your valuable suggestion. In accordance with your suggestions, we have updated the Materials and Methods section to better articulate the direction and strength of the correlations in our tables. The revised text is in line 90. This nuanced approach to arrow assignment allows for a more detailed representation of the relationships between biomarkers and cognitive outcomes in Alzheimer’s disease, providing valuable insights for both research and clinical contexts.

Accordingly, we have modified two tables to include a legend and arrows, with ‘two arrows indicating correlations defined by larger sample sizes or multivariate analyses, and one arrow for correlations identified through univariate analysis or in studies with smaller sample sizes.’

Comments on the Quality of English Language

This paper needs extensive language editing. Some of the sentences are extremely verbose and there are instances of the wrong verb being used. There is an excessive use of prepositions that make the manuscript feel like a listing rather than a literature review. I feel like speakers of other languages will have a hard time reading this review. 

Thank you for your suggestion. We have revised the manuscript to address concerns about verbosity, verb usage, and overall readability. We appreciate your input and welcome any further suggestions.

Reviewer 3 Report

Comments and Suggestions for Authors

The review about blood based biomarkers of Alzheimer's disease and cognition", is clearly presented and well organized. I just have some minor points to address: 

1) The methods, in particular the ultrasensitive SIMOA technology which allows the detection of low protein amounts in blood can be described briefly.

2) I would also discuss whether the blood-based diagnostic might replace the CSF diagnostic in future.

3) I would also give a brief summary or overview for the readers, either as a figure or a table showing the regulation of relevant AD related markers in blood, moderatly up-or downregulated, highly, etc. 

Author Response

The review about blood based biomarkers of Alzheimer's disease and cognition", is clearly presented and well organized. I just have some minor points to address:

1) The methods, in particular the ultrasensitive SIMOA technology which allows the detection of low protein amounts in blood can be described briefly.

Thank you for your insightful suggestion regarding the inclusion of a brief description of the ultrasensitive Single Molecule Array (SIMOA) technology in our methods section. We recognize the importance of this technology in enhancing the detection of low protein amounts in blood, which is crucial for the accurate measurement of biomarkers in Alzheimer's disease. Following your recommendation, we have updated our "Materials and Methods" section to include a description of the SIMOA technology (line 71): “Among these platforms is SIMOA (Single Molecule Array, Quanterix Corporation), a digital immunoassay technique that enables the detection of biomarkers at extremely low concentrations in blood samples. This technology is particularly significant for Alzheimer's disease research, as it allows for the accurate measurement of neurological biomarkers present in minute quantities, thereby facilitating early detection and monitoring of disease progression.”

In addition, we have introduced a sentence in the discussion section (line 402): “Notably, advancements in technologies like SIMOA have shown potential in overcoming these challenges. SIMOA's high-sensitivity detection capabilities make it a promising tool for accurately measuring neurological biomarkers in blood, which may facilitate early detection and monitoring of AD progression.”

2) I would also discuss whether the blood-based diagnostic might replace the CSF diagnostic in future.

Thank you for your valuable suggestion to discuss the potential of blood-based diagnostics in replacing cerebrospinal fluid (CSF) diagnostics in the future. This is indeed a pivotal area of discussion in the field of Alzheimer's disease research. We have incorporated a section in our discussion to address this important aspect, reflecting on the current state of blood-based diagnostics and their potential future role in comparison to CSF diagnostics (line 422): “In conclusion, this review represents a preliminary approach to the associations blood-based biomarkers for AD and cognitive measures While the potential for blood-based biomarkers for AD is promising, further extensive clinical validation is needed for routinary clinical application. The ongoing development in this area represents an exciting frontier in AD research, with the prospect of transforming diagnostic and treatment paradigms.”

3) I would also give a brief summary or overview for the readers, either as a figure or a table showing the regulation of relevant AD related markers in blood, moderatly up-or downregulated, highly, etc.

Thank you for your suggestion to provide a summary or overview of the regulation of relevant AD-related markers in blood. We recognize the importance of conveying this information in a clear and accessible manner. However, due to the varied methodologies employed in the studies we reviewed, we found it challenging to directly compare the strength of correlations across different studies.

As such, in accordance with your suggestion and to better articulate the direction of the correlations, we have updated the Materials and Methods section. The revised text is as follows (see line 90): ‘Presentation of Findings:

The findings are synthesized descriptively, highlighting the correlations between blood-based biomarkers and cognitive measures in the AD continuum. Summary tables (table 1 and 2) are included to provide a clear and concise overview of the observed associations of the main blood-based AD biomarkers: plasma Aβ and p-Tau and t-Tau measures. To enhance the clarity and accuracy of our review, we have implemented a systematic approach for assigning directional arrows in the correlation tables, which reflect the strength and nature of the correlations reported in the studies. This evaluation process was meticulously carried out by two senior neurologists and one neuropsychologist, ensuring a thorough and expert analysis of each study's findings. A single upward arrow (↑) indicates a positive correlation identified through univariate analysis or in studies with smaller sample sizes. Two upward arrows (↑↑) denote a stronger positive correlation, typically observed in studies with larger sample sizes or multivariate analyses. For negative correlations, where higher biomarker levels correspond to poorer cognitive performance or faster cognitive decline, a single downward arrow (↓) is used for univariate analyses or smaller studies, and two downward arrows (↓↓) are employed for stronger negative correlations identified in larger sample sizes or multivariate analyses.’

In line with your suggestion, we have modified two tables to include a legend and arrows: ‘two arrows indicating correlations defined by larger sample sizes or multivariate analyses, and one arrow for correlations identified through univariate analysis or in studies with smaller sample sizes.’

These changes aim to succinctly represent the methodological differences among the studies, focusing on whether the studies employed multivariate adjustments and the relative sizes of their sample populations. This methodological representation using arrows allows for a quick and clear understanding of the study design and analytical approach, providing a concise overview of their differences as you suggested.

Round 2

Reviewer 1 Report

Comments and Suggestions for Authors

Authors adequately addressed the critiques